# Barriers and Facilitators to Father’s Engagement in a Depression and Alcohol Use Intervention in Kenya: Father, Family, and Community Factors

**DOI:** 10.3390/ijerph20064830

**Published:** 2023-03-09

**Authors:** Ali Giusto, Florence Jaguga, Victor Pereira-Sanchez, Wilter Rono, Noah Triplett, Hani Rukh-E-Qamar, Mattea Parker, Milton L. Wainberg

**Affiliations:** 1Department of Psychiatry, Columbia University Irving Medical Center, New York State Psychiatric Institute, New York, NY 10032, USA; 2Department of Mental Health, Moi Teaching and Referral Hospital, Eldoret P.O. Box 3-30100, Kenya; 3Department of Child and Adolescent Psychiatry, New York University Grossman School of Medicine, New York, NY 10016, USA; 4Department of Psychology, University of Washington, Seattle, WA 98195, USA; 5Department of Psychology, McGill University, Montreal, QC H3A 0G4, Canada; 6Department of Psychology, Tuskegee University, Tuskegee, AL 36088, USA

**Keywords:** implementation, fathers, CFIR, sustainability, depression, alcohol use, family

## Abstract

In Kenya, there is a treatment gap for depression and alcohol use that is especially large for fathers, which has consequences for families. While treatments exist, there are challenges to implementation. This study aimed to understand barriers and facilitators to implementing a treatment for fathers’ depression and alcohol use in Eldoret, Kenya. Guided by the Consolidated Framework for Implementation Research and the Integrated Sustainability Framework, we conducted 18 key informant interviews and 7 focus group discussions (31 total participants) with stakeholders in Eldoret (hospital leaders, policy makers, mental health providers, community leaders, fathers, lay providers, and patients previously engaged in treatment). Interviews were analyzed using the framework method; themes were matrixed by framework domains. Participants identified barriers and facilitators, and opportunities for implementation, in the following domains: innovation, outer setting, inner setting, individual, sustainability, and characteristics of systems. Barriers included a lack of resources, stigma, masculine norms, cost of services, and alcohol dependence. Facilitators included community buy-in, family support, providers with lived experience, government support, and relevant treatment content. Findings will inform implementation strategy development for an intervention for fathers with depression and alcohol use, and family problems with local relevance and scalable potential.

## 1. Introduction

Depression is the leading cause of disability worldwide [1] and is often comorbid with alcohol use [2]. It disproportionately impacts men and accounts for 9.6% of disability adjusted life years [3,4]. Men are also unlikely to seek mental health treatment, complicating the burden of illness [5], with men initiating and staying in treatment at rates four times lower than women [6,7]. Further, the consequences of men’s depression and alcohol use often extend to families, who are at higher risk for violence, poor relationships, and mental health issues [8]. Despite these consequences, there is an estimated 78% treatment gap for alcohol use disorders (AUDs; mental health disorders characterized by an impaired ability to stop or control alcohol use despite adverse social, occupational, or health consequences [9]), and 75% gap for depression in low-to-middle income countries (LMICs) [10,11]. This is largely owing to a scarcity of resources and treatments targeting these issues and family-related problems [12]. 

These problems are reflected in Kenya, where prevalence of AUDs is 7 times higher in men than women (7.1% men, 0.9% women) [13]. In Kenya, treatment is limited to a small number of counseling and rehabilitation centers [14], often serving only severe cases with limited capacity and relatively high cost. Substance use disorder treatment is only available in three publicly supported clinics across Kenya, all of which are located in urban centers [14]. There is a national protocol for treatment of substance use disorders in Kenya that calls for brief intervention and a combination of pharmacological and psychosocial interventions in inpatient rehabilitation centers [15]; however, little is known about the implementation of these interventions in different settings. Although efforts have been made to expand services, treatment seeking and awareness remain low [16]. Further, similar to other men globally, Kenyan men have low treatment seeking and engagement [5,17]. Therefore, there is a need for interventions that address alcohol use and depression, engage men, and consider delivery for resource scarce settings.

In response to the lack of services to address depression, unhealthy alcohol use and related family problems, a team of Kenyan and US-based clinicians and researchers developed an intervention for fathers in Kenya called ‘Learn, Engage, Act, Dedicate’ (LEAD) [18]. LEAD is a five session intervention anchored in motivational interviewing (MI) and behavioral activation (BA) delivered by lay providers who are peer fathers [18]. LEAD integrates family-related material explicitly and addresses central cultural factors through discussions of masculinity [18]. LEAD begins with MI-informed strategies to build readiness to change, engage men in treatment, and establish commitment [18]. In a proof-of-concept trial, LEAD improved men’s drinking, with men 5.1 times more likely not to drink on a given day post-treatment, and consuming 50% less alcohol on days they did drink [19]. Pre-post 1-month survey results also showed improvements in fathers’ parent–child and couples relationship quality and family involvement, and individual mental health reported on by men, their partners, and one child with medium to large effects [19]. Lay provider delivery also showed initial acceptability according to providers’ qualitative interviews and high rates of intervention fidelity (M = 93.8%) and general clinical competency during delivery (M = 3.2; moderate to optimal) [20]. 

However, we have yet to examine how to best work with community strengths to deliver LEAD sustainably and with potential for scale. Here, we aimed to understand how LEAD and interventions for father mental health can be implemented within Eldoret, Kenya in ways that consider sustainability and scalability. Eldoret was chosen because our team has a long-standing relationship with partners in Eldoret, and Eldoret was selected as the location in which the LEAD intervention could be scaled. The study also aims to explore barriers and facilitators to reaching, engaging, and retaining men in mental health services in the area to understand key factors for father treatment engagement. 

## 2. Methods

### 2.1. Overview and Guiding Frameworks

To conduct the study, we were guided by two implementation determinant frameworks, the Consolidated Framework for Implementation Research (CFIR) and Integrated Sustainability Framework (ISF) [21,22]. The identification of these determinants factors has the potential to help reduce the burden of depression and alcohol in the area, build capacity, and later delineate sustainable and locally relevant implementation strategies that may be relevant for hard-to-reach populations, such as men.

To identify implementation determinants, we used a qualitative approach. The Consolidated Framework for Implementation Research (CFIR; [21]) and Integrated Sustainability Framework (ISF; [22]) guided assessment and interview guides. We chose CFIR (CFIR; [21]) and Integrated Sustainability Framework (ISF; [22]) because it is an implementation determinant framework that provides a practical guide for identifying barriers and facilitators of implementation across five domains. Domains are: (1) innovation characteristics (intervention features); (2) outer setting (economic, political, and social context); (3) inner setting (organizational structure and culture); (4) characteristics of individuals (beliefs, attitudes, and behaviors); and (5) implementation process (planning, engaging, and executing). The ISF was included given its overlap with CFIR (CFIR; [21]) and Integrated Sustainability Framework (ISF; [22]) and additional emphasis on the construct of sustainability (continued health impact and capacity building). As such, these domains were assessed (5 CFIR + ISF sustainability) through key informant interviews and focus group discussions with diverse stakeholders in the area. Interview guides also included questions on participants perceptions of the problem, the consequences of the issues, strategies for addressing barriers, and prompts to provide examples of successfully implemented programs with men.

### 2.2. Setting

Research was conducted in Eldoret, Kenya, in collaboration with Moi Teaching and Referral Hospital (MTRH), and AMPATH, a service and research organization including MTRH and a consortium of North American schools in partnership with the Kenyan Ministry of Health. All study procedures were approved by the Institutional Research and Ethics Committee at Moi University/Moi Teaching and Referral Hospital and the Institutional Review Board at New York State Psychiatric Institute. 

### 2.3. Participants 

We conducted 18 individual key informant interviews (KII) and 7 focus group discussions (FGDs). KIIs were completed with four community leaders; four policy makers; four hospital leaders; five professional alcohol use and mental health providers; three previous LEAD trial participants; and three previous pilot trial lay providers (i.e., peer fathers with no previous counseling training). FGDs were completed with fathers with problem drinking, mental health providers (2), and community leaders. See Table 1 for details. Across interviews and groups, for providers, we aimed to sample individuals from the private and public sectors; from rural and urban areas; and lay and professional providers. We identified and recruited leaders and providers from four catchment areas in and surrounding Eldoret. For leaders, we identified individuals’ representative of the community based on gender, role, and religion. Women participated in both KIIs and FGDs, with one FGD being comprised exclusively of women community leaders. Age was not collected across all interview types (e.g., we did not collect age from hospital leader or policymaker participants). For both the KII’s and FGDs, on the ground partners and Kenyan research staff identified leaders based on previous knowledge of communities and past project experiences with leaders. 

#### Recruitment 

A trained Kenyan research assistant (RA) or the Project Coordinator (PC) approached leaders either by phone, email, or in person to briefly describe the study and assess initial interest. If the leader expressed interest, the RA or project coordinator completed consent. Previous patients and providers were identified through project records if they expressed interest to be contacted in the future about the study. If so, the trained Kenyan RA called past providers to briefly describe the study and assess initial interest. If they expressed interest, the RA set a time to review and obtain consent in person. All three providers from the previous study were contacted. For previous participants, their identification numbers were randomly assigned a number 1–9 using a Microsoft Excel random number generator. Those assigned 1–4 were first invited to participate, followed by 5–9 if the initial subjects declined. The RA called past participants or met with them in person to briefly describe the study and evaluate initial interest. If interest is expressed, the RA obtained consent in person.

For current patient participants, they were considered eligible if they were fathers (i.e., responsible for the caregiving of at least one child younger than 18), reported problem drinking as indicated by a score of 8–19 on the Alcohol Use Disorder Identification Test (AUDIT; a widely used 10-item screening tool developed by the World Health Organization [23]). The AUDIT assesses frequency of alcohol consumption, volume of consumption, and perceived consequences of alcohol consumption. Scores from 8 to 14 on the AUDIT suggest “hazardous or harmful alcohol consumption,” and scores 15 or greater suggest likely alcohol dependence [23]. Men must have also reported alcohol use within the past 2 months to be eligible. Given the scope of this study, we did not gather comprehensive information on the men’s history of alcohol use or other mental health problems. Patients were men presenting for initial evaluation of alcohol use or general mental health problems at MTRH and men attending MTRH affiliated monthly alcohol use support groups. These men were approached by an RA, who explained the study, and if initial interest was expressed, screening was completed. If eligible, men were consented. If eligibility was not met, men were assured that they would continue receiving care from MTRH and that their services would be unaffected. 

### 2.4. Procedures

Trained Kenyan research assistants conducted data collection, and all FGDs and KIIs in conjunction with the onsite investigator. All guides were available in both English and Kiswahili based on the participants preferences. KII and FGD guides are included as Appendix A. All interviews and FGDs were audio recorded on secure devices, deidentified, and transcribed into English by an RA with extensive experience transcribing in both Kiswahili and English. The RA was a native Kiswahili speaker who had been trained in transcription, including translating verbatim and retaining any Kiswahili terms that may not directly translate into English. 

### 2.5. Analysis 

Qualitative data analysis was guided by the framework method [24]. We used both an inductive and deductive process, balancing data—driven codes with a priori framework codes. The analysis team included a Kenyan psychiatrist and clinical researcher, a Kenyan research manager with extensive mental health research experience, a US-based psychologist with previous experience working in Eldoret, a US-based undergraduate-level research assistant, and a US-based psychiatrist. 

Analysis followed seven general streps. Step 1 consisted of interview transcription (English with specific Kiswahili idioms maintained as appropriate). Step 2: the team read and familiarized themselves with the transcripts, selecting a few interviews from each participant type to review. General annotations and ideas were memoed, then discussed in a team to inform development of codes alongside review of relevant CFIR and ISF domains. Step 3: a random sample of transcripts from each interview type (i.e., both KII and FGD transcripts) were divided and coded using inductive codes. Next, for Step 4, in group discussions, the team developed the coding framework and codebook using inductive and deductive codes. We then independently coded transcripts and refined the codebook until 80% agreement was met across four coders (FJ, WR, VP-S, and AG). [Note: we opted for percent agreement because it was unlikely that coders would be “guessing,” and one advantage of Kappa is to account for chance agreement. Additionally, percent agreement is directly interpretable [25]]. In Step 5, after agreement was reached, transcripts were divided and coded independently; questions that arose were discussed and consensus around the question were reached. Transcripts were coded using NVivo 12.0 software. Step 6: coded data were organized by interviewee type and reviewed. Lastly, in Step 7, the team interpreted data. We did this through synthesis and summary and re-organizing data by specific CFIR and ISF constructs, and codes in a matrix. When exploring CFIR constructs, we were also guided by a review of the CFIR in LMIC, which suggests an additional construct—“Characteristics of Systems”—to increase the CFIR’s compatibility with LMIC settings [26]. ‘Characteristics of Systems’ relates to the relationship between key systems characteristics and implementation. Additionally, after the completed of data collection, an updated CFIR framework was published [27]; therefore, we used these updated domains in our final step of coding. 

## 3. Results

We present barriers and facilitators to care within each framework domain. Within these domains, we also present opportunities for successful implementation and engagement that emerged from the data. Opportunities typically constituted suggestions for leveraging strengths or adding elements to a domain that could enhance implementation or engagement. These were distinct, albeit overlapping, with determinant-level facilitator.

### 3.1. Overview

Table 2 presents the barriers, facilitators, and opportunities that emerged by implementation framework domains (CFIR/ISF) (CFIR; [21]) and Integrated Sustainability Framework (ISF; [22]). Key CFIR/ISF domains across the data included those at the following levels: innovations, outer setting, inner setting, individual, characteristics of the system, and sustainability (CFIR; [21] and Integrated Sustainability Framework (ISF; [22]). No clear themes emerged in the domain of implementation processes; as such, this domain is not reflected in Table.

### 3.2. Innovation-Level

Within the innovation domain (i.e., perceptions of the intervention itself), facilitators were noted across participants and interviews. Participants emphasized keeping the intervention short, specifically five sessions or less. Some community and hospital leaders also reported a short intervention could fit within a stepped-care approach in which men complete treatment, then those needing higher care are referred to inpatient treatment or ‘step down’ to more infrequent group—sessions in the community. This was seen as an opportunity. Many noted that it may be important to incorporate the lived experience of peers experiencing problem drinking or depression. Mental health providers noted that this approach could get more men to come to the program and stay in it. The cultural relevance of the treatment was also noted as a facilitator, such as emphasizing a focus on leadership (e.g., engaging in this program can help one to be a better leader or provider). One mental health provider, a psychiatrist, noted the following:


*The program sounds very good because of this good leader and good father aspect in it. This are the things that we need to teach this man that is growing up. You see a role model who has gone through it has become role model despite whether he was taught or not but has become a good father and a good leader. So, we are instilling those values into now this man again.*
—Hospital Psychiatrist

Another mental health provider reported, “*I would say that by nature and culturally, men are supposed to be leaders. And because they are leaders, they have been seen to be socially the ones who are supposed to offer help, they have the solutions for every good*.” Participants noted mixed views on group therapy as both a facilitator and a barrier, saying groups can enhance peer support, yet privacy and confidentiality may be a concern. Although some previous program providers noted including family members in treatment as a facilitator, most participants, including current men experiencing issues with drinking, saw this as a barrier due to privacy concerns and stigma. Lastly, one opportunity that was emphasized was the potential inclusion of support groups following the program to support sustainability of any change.

### 3.3. Outer Setting

At the outer setting level, which includes community, county, or country-level factors and attitudes, different barriers, facilitators, and opportunities emerged. Cultural norms were cited as both barriers and facilitators. A few participants noted that in some cultures (based in tribal ethnicity) alcohol use is not allowed, which may facilitate improved alcohol use outcomes. Conversely, in some cultures within Kenya alcohol use may be tied to celebrations and traditions, such as circumcision ceremonies. Alcohol use was also noted as a stigmatized problem in the community, which was a barrier. 

Norms around alcohol use and mental health stigma often overlapped with norms around masculinity and what it means to be a man in the community. Almost all interviewees noted masculine norms as a barrier to help-seeking and treatment engagement. Most participants specifically noted that norms around being a man discouraged help seeking. Statements often reflected the following: “*In my culture, you do not ask for help. Once you are out of there you are a man and there is no way you can go and ask for help from your fellow man. You will fight on your own as a man*.” Similarly, a hospital leader said, “*You know I am a [cultural group] man and I am not supposed to be taken to a rehab. I am supposed to fight my own fight*”. Stigma and cultural norm barriers appeared to go hand in hand. One policy maker summarized the ways in which multiple stigmas surrounding alcohol use, masculine norms, and alcohol use can impact men’s help seeking:


*There is the cultural aspect where the men have [themselves] together. Man is not supposed to say that I am overwhelmed, so the culture is assuming that the man has to have himself together. Two, men are known to drink, culturally, you are not supposed to say that I cannot drink. Three, there is the fear of failure. The stigma associated with it, if want to seek help for mental health, there is already the stigma of that you are weak. You are crazy and some sort. So, someone may know that they need help but they are afraid on how the society will view them.*
—Policy Maker

Communities (i.e., systems outside hospital/primary care settings) and families were emphasized as critical facilitators. Communities were seen as key to supporting fathers and their families, and for delivering treatment. Families and communities were also often noted as potential opportunity sources for referring and engaging fathers in care, help-seeking, and making change. Related to families, a current patient noted, “*You know when you have a wife, the wife pushes you slowly by slowly to stop alcohol*”. In a focus group with mental health providers, one provider also noted the importance of family in help seeking, saying:


*Mostly they do not seek help [unless] they have a strong person in the family to consider them and even persuade them to seek help. Like now in a family, if the husband result to a lot alcoholism, the wife will have to talk to the family members, talk to the closest friends and if they have resources to meet the expenses for the rehabilitation and the rest, they will go for it.*
—Mental Health Nurse Counselor

Many participants cited the community as being necessary for successful implementation. One policy maker summed up this opportunity clearly while also noting some of the cultural aspects of community as facilitators. The policy maker noted the following: 


*I think the resource that we have in our country is the community itself. The way we are arranged in our neighborhood, the families are close to each other, they are aware of each other, and we live in a communal way so that we have adapted that culture of if somebody’s child is struggling with this problem, it is my problem too. That kind of arrangement is a big resource. That is why I am saying that using the community, as the entry point could be the best approach ever.*
—National Policy Maker

Leaders within the community, including religious leaders, also represented sources of opportunity to support implementation. Participants noted that community and religious leaders could help facilitate referrals into the program given the fact that men may often naturally come to them. One participant noted: “*There are those who can be identified by their religious communities when someone has been omitting their religious obligations or religious functions*”. Another participant commented on community stakeholders saying: “*If you want your program to get right down to the village, then the community health volunteers are vocal persons, those are important people. They will help you to realize mass success in the program*”. Many participants also emphasized the importance of leaders being friendly in approaching fathers to succeed. This was especially true of community leaders many of whom noted sentiments similar to this village leader, who said: “*Be friends with them but if you go with in a harsh manner, you will not succeed*.”

### 3.4. Inner Setting

The inner setting domain compromises the setting in which the program will be delivered. In this case, the inner setting included either a community-setting or a hospital setting. Given the delivery setting for the program had not been previously determined, results in this domain typically focused on setting opportunities to consider in program delivery with some noted barriers and facilitators to these settings and existing services in the area.

When discussing hospitals or rehabilitation centers for delivery, the primary facilitator was having trained professionals and access to medications. At the hospital, access to other health services was noted as a facilitator. Hospital barriers included stigma around seeking care within the hospital specifically, distance from some communities, and cost. The largest rehabilitation center barrier was high costs, long waitlists, and a scarcity of centers. At the community-level, a noted barrier was a “shortage of workers, lack of counselors” in the community to deliver care. A facilitator was an overall sense of belonging, which could support change. Inner setting opportunities for delivery included the suggestions to deliver the program far from areas and places that may induce urges to drink, such as bars. Additionally, participants reported delivery in communities should occur in accessible locations, such as community centers. 

### 3.5. Individual-Level: Participants and Providers

The individual-level domain consists of perceptions of individuals including their roles and characteristics. Results focused on provider and participant factors. At the participant/patient level, facilitators to successful delivery included whether men self-refer, are motivated (to both change and take care of their family) and have friends who support them. Almost all participants emphasized severe alcohol dependence and/or polysubstance use as a barrier. Dependence and other drug use were cited as barriers to engaging and retaining men, and their potential success in treatment. 

At the provider level, facilitators included certain characteristics, such as not being known to the participant, but being from the community, having a friendly warm manner, and being conscientious (e.g., being on time and effective). Barriers included fear of doctors in medical settings, a point that current patient interviewees emphasized, and skepticism of individuals outside of the community. A potential opportunity was the possibility of including perspectives of community members in recovery. For instance, one participant noted the following: “*We should be having mentor fathers [providers] who have undergone certain behavior change [and] use them as advocates*”. Another, a hospital leader, said, “*The best way of helping them [men] is using those who have already recovered, it has really worked, that is the link*”. Others noted opportunities to work with providers in communities who have some existing mental health training or status in the community. 

### 3.6. Sustainability (ISF) 

All aspects of the program related to sustainability focused on potential opportunities for fostering sustainability. Sustainability referring to the continued impact and delivery of an intervention over a long period of time [28]. Opportunities included the following: incorporating county-level officials and policy makers early in implementation, integrating services into existing structures, determining a long-term funding source (likely governmental), training a broader work force, incorporating sensitization to mental health in the community and primary education, community perceptions that the program works, community buy-in, and program champions who will advocate when the program is done. 

Related to working with policy makers, one interviewee said: “*politics influence everything and our politicians direct the way things go. If we have politicians champion these things, we will have a better outcome; for you to enter the community you have to get the key person who is the political figure head there*”. Similarly, a hospital leader noted:


*Number one, any program that is going to be started now, for example for mental health, you need to incorporate the county, which will be owned by the county itself. This is because if it is owned by the county and we are just supporting the county to push for that. For example, our governor, he is very good at fighting for those mental issues.*


Regarding sustainable funding and politics, one policy maker expressed an opportunity for there to be more transparent communication and execution of policies that are meant to repurpose money taxed for alcohol sales back into treatment. They noted that these policies exist, but are rarely implemented and not widely known. One policy maker suggested having patient’s pay for such a program to build out a long-term infrastructure until services might be reimbursable through insurance. A mental health provider commented that not including incentives for project participation could be a long-term approach for helping with sustainability. 

While most noted how critical funding is, a few noted the importance of training with one participant saying: “*we could have the funds but if the manpower is lacking and competence, then the funds will go to other things. People will purport to support mental health, but they are not trained in it*”. Many participants mentioned the importance of community buy-in and “social mobilization.” Important to community buy-in were opportunities to increase general community awareness of mental health and the inclusion of such topics in school. Similarly, a few participants noted that others in the community seeing change in fathers from the program could help to ensure its sustainability over time. One hospital leader further noted the importance of advocates remarking the following: 


*The other thing is to get champions, what I have seen with my little experience is to get people on the ground who can advocate. Whether they are patients or health care workers, who after the program is done, they are willing to keep fighting for it. It is not enough to train people; it is enough to get a champion, someone who is very passionate about such a program. Because otherwise when the turbulence of life happens and maybe there is low workload, less payment or whatever else, the program is likely to die if there is no one who is continuously financing for it.*


### 3.7. Characteristics of Systems (CFIR-LMIC) 

Characteristics of systems describes the relationships between systems and implementation. In this domain, a few barriers and facilitators emerged. Barriers included limited existing services and difficulties linking men to other levels of care, limited resources and providers, and inequitable policies. Commenting on limited services, especially for individuals who have fewer resources, a hospital leader noted: 


*In terms of linking the men to care, there is a real gap. One, mostly the ones that I know [who receive care] are people who are known, but [for] those who come from less privileged family backgrounds, it will be very difficult cater for this individuals… the ones living in slums they are only arrested, they are threatened and told that if you continue with this alcohol or this and this, we find you next time we will charge you like this. It is like enforcement of behavior change.*


Another participant noted: “*I feel that the resources are not sufficient, even the staff who work in the primary health care do not understand how to manage patients with mental illness. They need to be retrained, they need a lot of training and exposure, so I think human resource is a challenge*.” 

One facilitator noted in this area was combined religious and inpatient treatment centers in part because these types of facilities are often run by individuals who have past mental health or alcohol use lived experience. The participant noted: “*When you look at the spiritual based centers, majority of the owners are the one who were substances users and they have changed a lot of lives through that. So having that around actually makes, them [men] think about help because they want to be like them*.” One opportunity noted by a few participants was the potential for a stepped model for implementation using communities for referral and long-term outpatient care with referrals to hospitals for more moderate to severe concerns. 

## 4. Discussion

To our knowledge, this is the first study to apply an implementation framework to examine barriers and facilitators of delivering an intervention for fathers in Kenya. This study explored determinants influencing implementation of an intervention for depression, alcohol use, and family engagement called ‘Learn, Engage, Act, Dedicate (LEAD)’ from the perspectives of multiple Kenyan stakeholders. Participants identified several barriers and facilitators within innovation, outer setting, inner setting, individual, sustainability, and system characteristics domains of the CFIR plus the CFIR adapted for LMIC and a sustainability domain. Opportunities for capitalizing on facilitators or mitigating barriers also emerged from interviews. 

Barriers and facilitators identified in this study are relatively consistent with a recent systematic review of determinants to implementing adult mental health services in LMIC primary care settings [29]. For example, research from diverse LMIC have similarly demonstrated stigma and lack of resources as barriers, similar to factors noted here. Unique in our study are determinants that may be particularly important when considering interventions for men and fathers and those in community settings. For example, in this case, community members and families were seen as essential facilitators for implementing treatment. Community members and families were also seen as sources of opportunity to enhance father engagement in part through recruitment. This aligns with previous qualitative work in the area indicating families and community members are often the first and only source of help that men experiencing problem drinking or mental health issues accepted [30]. As such, working with community leaders and family members may be an effective implementation strategy to be tested in the future [20].

Another barrier unique to this study and population were masculine norms. Norms around what it means to be a man were perceived as critical impediments to men’s engagement and retention in care. This is consistent with a broader literature focused men’s mental health which has found hegemonic masculinity to interfere with men’s mental health engagement [6,7]. (Hegemonic masculinity often relate to expectations for men to be invulnerable and self-sufficient [31].) Masculinity, or more broadly gender norms, have been explored less often within the field of implementation science. Yet, one study exploring barriers and facilitators to implementing treatment for alcohol use and intimate partner violence in Mozambique did report that those interviewed had concerns about men’s engagement in such treatment due to traditional masculine norms [32]. This suggests future implementation strategies might consider and address gender norms and their intersections with stigma. Strategies might focus on outer setting factors, such as radio campaigns, or intervention level elements, such as tailoring content to men [18,33]. To address these factors in our work, we included content related to traditional gender norms in our intervention. We also used peer fathers as providers to facilitate men’s participation, and model male help-seeking and provisions for participants. 

A study strength was the application of a systematic implementation framework and the inclusion of sustainability as a domain. Consideration of potential sustainability factors is especially important to intervention design and implementation in low-resource settings. If an innovation or its delivery approach is not sustainable within a setting, it offsets its potential benefit within a community [34]. Not considering sustainability may also exacerbate disparities and undermine community trust in research. For instance, if a project comes into a community then leaves when the funding is expired, it can foster mistrust, leave gaps in care, and be perceived as taking advantage of the community [35]. Here, we tried to consider elements that may impact potential for sustainability to guide later development of implementation strategies alongside the piloting of the treatment. This led to the identification of opportunities within the data here that can work with community strengths, such as incorporating policy makers early in implementation. These findings have also motivated our continued engagement with community leaders and policymakers as we begin to scale our intervention. For example, we intend to include policymakers as advisory board members to plan for later funding and sustainment. 

Our findings should be interpreted within the limitations of the study. First, our sample, while diverse, was limited to individuals in the Eldoret area. This is appropriate for the scope of the study but limits generalizability to this area. Next steps might include a sampling from broader areas. Related, family members of individuals experiencing depression or alcohol use were not interviewed. Inclusion of their perspectives in future work could elucidate other determinants and opportunities. Furthermore, when we originally developed the interview guide, we did not include the additional CFIR-LMIC domain of system characteristics. Although themes within this domain still emerged it is possible deeper insight would have been gained with its specific inclusion in guides. Lastly, although guides focused on both depression and alcohol use, participants primarily focused on alcohol use limiting knowledge about depression. 

## 5. Conclusions 

Guided by the Consolidated Framework of Implementation Research and the Integrated Sustainability Framework, we found barriers, facilitators, and opportunities to delivering a mental health and alcohol use intervention for fathers in Eldoret, Kenya. Factors fell within domains of innovation, outer setting, inner setting, individual, sustainability, and characteristics of systems. To our knowledge, this is one of the first studies to explore determinants influencing fathers’ engagement and retention in a mental health and alcohol use intervention in Kenya using systematic implementation frameworks. Next, cultural and contextual factors identified here will inform the mapping of implementation strategies. Findings can also inform a broader understanding of key factors requiring consideration to engage fathers in mental health care, an essential ingredient for family well-being. 

## Figures and Tables

**Table 1 ijerph-20-04830-t001:** Participant Interview Types.

Interviewee	Interview Type	Number Conducted (# Participants)
Community Leaders	KII	4 Interviews (4 participants)
Policy Makers	KII	3 Interviews (3 participants)
Hospital Leaders	KII	5 Interviews (5 participants)
Past LEAD Providers	KII	3 Interviews (3 participants)
Past LEAD Patients	KII	3 Interviews (3 participants)
Community Leaders	FGD	2 Groups (4 participants)
MH and AUD Providers	FGD	3 Groups (21 participants)
Current Patients	FGD	2 Groups (6 participants)
Total # Interviews/Participants	—	KII: 18 interviews (18 participants) FGD: 7 Groups (31 participants)

Note: MH = Mental Health; AUD = Alcohol Use Disorder; LEAD = Learn, Engage, Act, Dedicate; KII = Key Informant Interview; FGD = Focus Group Discussion.

**Table 2 ijerph-20-04830-t002:** Implementation Barriers, Facilitators, and Opportunities by CFIR/ISF Domains.

CFIR/ISF Domain	Implementation Barriers (−), Facilitators (+), and Opportunities (*)	
INNOVATION	+ Short time commitments (≤5 sessions) + Intervention integrates peer support + Intervention highlights culturally relevant values, such as leadership ± Group-based format facilitates peer support, but is less private - Recording sessions may be uncomfortable for participants - Some men may not wish to incorporate their families in treatment * Introducing support groups at the end of the intervention to improve sustainability	
OUTER SETTING	+ Communities and families support each other ± Cultural norms surrounding alcohol use - Cultural norms surrounding masculinity and stigma may impact engagement with treatment - Lack of financial resources * Family and community members may support with referral and engagement * Connecting with community and religious leaders to support program implementation	
INNER SETTING	* Intervention should be delivered far from alcohol use locations * Intervention could be delivered in more accessible locations within communities, such as community centers	
INDIVIDUAL	Participants + Men who self-refer may be more likely to succeed - Men may be impacted by severe alcohol dependence and/or polysubstance use	Providers - Fear of doctors in medical settings - Skepticism of individuals from outside the community * Incorporate perspectives of community members in recovery * Providers should have some training and status in community
SUSTAINABILITY	* Incorporating county-level officials and policymakers early in implementation * Integrate services into other community structures, such as churches, where individuals may expect to volunteer and support each other * Incorporate content into primary education as a preventive measure	
CHARACTERISTICS OF SYSTEMS (CFIR-LMIC)	+ Existing spirituality-based rehabilitation programs - Limited existing services and difficulties linking men to care - Limited providers and resources to support new interventions - Lack of equitable policies * Linking community and hospital services	

Note: - = barriers; + = facilitators; ± = barrier and facilitator; * = opportunities for positive engagement/implementation; CFIR = Consolidated Framework for Implementation Research; ISF = Integrated sustainability framework

## Data Availability

Data may be available upon request (ali.giusto@nyspi.columbia.edu).

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
