# Peer review of "Barriers and Facilitators to Father’s Engagement in a Depression and Alcohol Use Intervention in Kenya: Father, Family, and Community Factors"

_ijerph, 2023, doi:10.3390/ijerph20064830_

Round 1

Reviewer 1 Report

Dear Authors, 

Thank you for your effort and energy in doing this study. I consider that it is interesting and well-described. 

I want to make some observations that you can consider to improve your manuscript.

Line: 110-122, you described your participant's group nicely. But in line 120, I am not clear about the gender (where all male?), and the age of the participants would be essential to see more clearly the group characteristics. 

Table 1. On the same table, put in sequence the participants that served as KII and participants that served as FGD. This will help to do an easy reader Table 1. Please clarify, the number of interviews does not coincide with the number of participants. I understand that you did interviews, but in your description, you mention that for all participants, you did interviews. Then the table and the report are a little confusing. 

In line 157 analysis, you mention that after the transcriptions agreement, you used codes and estimated a percentage. Did you do this for all interviews, leaders, providers, and patients?   

Table 2. Please correct "sustianability"

Table 2. In column "individual," you mention that men are impacted by severe alcohol dependence or polysubstance use. Do you discriminate between these two different conditions?

Do you have any idea about when the participants started drinking alcohol? This would be important to have a more exact idea about the level of dependence between one single addiction to a mixture. 

Thanks for considering this journal. 

Sincerely

Author Response

Please see the attached file for our responses to reviews by both reviewers. Again, we thank you for your time and energy reviewing this work.

Reviewer 2 Report

Introduction:

The introduction of the manuscript begins with a lengthy background on the disease topic. In order to convey the message appropriately, the first sentence must be broken down into multiple sentences. For example, create one sentence to describe the disability associated with depression, another sentence to provide more geographic information on the statistics mentioned in the first sentence, and then a sentence to describe the adjust life years so the reader knows what this is related to and the geographical area the statistic came from. Towards the end of the first paragraph there is an abbreviation provided (LMIC), but no background information for what is stands for, therefore a new sentence could be created to provide that knowledge for the reader.

Another weakness of the introduction is the usage of alcohol use disorders, without background information on how the diagnosis was concluded such as from guidelines or the DSM5. Furthermore, the authors mention use of rehabilitation centers but fails to provide context as to what a rehabilitation in Kenya entails. The manuscript explains it use of “LEAD” throughout the third paragraph but fails reference the article each time it is mentioned. A significant concern is the implementation in Eldoret but fails to provide the reader any background knowledge on why Eldoret was chosen.

Finally, in the last few sentences from line 79-84 the author begins presenting the methods and should relocate this information to the methods section. The introduction must end with the objective of the study.

Methods:

There are some weaknesses in this section. For example, in line 89, the author failed to reference the domains of CFIR, although they did reference CFIR in the previous sentence. The domains must be referenced to avoid plagiarism. Another concern is the usage of Alcohol Use Disorder Test which was used by the authors to determine the inclusion and exclusion criteria because additional details are needed to understand how it was utilized. Is the Alcohol Use disorder test a universal tool to diagnosed? If so, the authors should state that. A significant concern of the methods is the absence of guidelines to diagnose and treat the population under investigation. Under the procedure section in the methods, the authors also need to provide more details on how the bias was avoided during the process of translation. Also, please provide interview guide and interview details such as the duration of the individual interviews and the focus groups and whether the same interview guides were used for both groups.

Results:

The author begins this section with an overview from the methods section to provide a easy transition to presentation of the results. The author fails to provide a reference for the different frameworks used in this article, these frameworks must be referenced. The author effectively sets up an easy-to-follow table at line 199 which includes a description and the table also include a easy to follow key to assist with interpretation that is categorized clearly by the domain section. Each section in the results the author provides a description of that specific domain which helps effectively establish the differences between the sections. The author distinctly provides direct quotes from those who were involved such a participants, psychiatrist, and policy makers to further demonstrate the domains result, but should consider separating into a new line and italicizing all quotes to have them appear more noticeable to the reader.

Discussion:

Overall, the section provides a good summary of the study's findings and their implications but providing more specific examples and information about the implications of the findings, and limitations of the study would make it even stronger. The authors compared the findings with previous studies with the same findings makes the discussion a strong statement. Also differentiating the findings which were newly found by the studies was also a great addition to the section. The limitation of the study was also well described.

Conclusion:

The section highlights that the study is one of the first to explore determinants influencing fathers’ engagement and retention in a mental health and alcohol use intervention in Kenya using systematic implementation frameworks, which is a strength of the study. Overall, this section provides a clear and concise summary of the study's main findings.

Author Response

Please see attachment for responses to both Reviewer 2 and Reviewer 1. Again, we thank you for your time and energy reviewing this manuscript. 

Round 2

Reviewer 2 Report

The authors addressed my suggestions.